# Dynamic Estimation of Saturation Flow Rate at Information-Rich Signalized Intersections

**Yi Wang** [1], **Jian Rong** [1], **Chenjing Zhou** [2],*** and **Yacong Gao** [1]

1    Beijing Key Laboratory of Traffic Engineering, Beijing Engineering Research Center of Urban Transport Operation Guarantee, Beijing University of Technology, Beijing 100124, China; wangyi330789@gmail.com (Y.W.); jrong@bjut.edu.cn (J.R.); gao17812102290@163.com (Y.G.)
2    School of Civil and Transportation Engineering, Beijing University of Civil Engineering and Architecture, Beijing 100044, China
*    Correspondence: zhouchenjing@bucea.edu.cn

**Abstract:** Intersections are the bottlenecks of the road network. The capacity of signalized intersections restricts the operation of the road network. Dynamic estimation of capacity is necessary for signalized intersections refined management. With the development of technology, more and more detectors were installed near the intersection. It had been the information-rich environment, which provided support for dynamic estimation of capacity. A dynamic estimation method for a saturation flow rate based on a neural network was developed. It would grasp the dynamic change of saturation flow rates and influencing factors. The measure data at three scenarios (through lanes, shared right-turn and through lanes, shared left-turn and through lanes) of signalized intersections in Beijing were taken as examples to validate the proposed method. Firstly, the traffic flow characteristics of the three scenarios and factors affecting the saturation flow rate were analyzed. Secondly, neural network models of the three scenarios were established. Then the hyperparameters of neural network models were determined. After training, the neural network structure and parameters were saved. Lastly, the test set data was validated by the training model. At the same time, the proposed method was compared with the Highway Capacity Manual (HCM) method and the statistical regression method. The results show that both regression models and neural network models have better accuracy than HCM models. In a simple scenario, the neural network models are not much different from the regression models. With the increase of complexity of scenarios, the advantages of neural network models are highlighted. In through-left lane and through-right lane scenarios, the estimated saturation flow rates used by the proposed method were 7.02%, 4.70%, respectively. In the complexity of traffic scenarios, the proposed method can estimate the saturation flow rate accurately and timely. The results could be used for signal control schemes optimizing and operation managing at signalized intersections subtly.

**Keywords:** traffic engineering; signalized intersections; dynamic estimation; neural network; saturation flow rate

---

## 1. Introduction

With the fast-growing stage of urbanization and mobility, the number of vehicles in developing countries has increased year by year. Traffic jams have become frequent in many cities. In Beijing, the transport annual report stated that the average daily congestion duration was 2 h and 50 min in 2018 [1]. Intersections are the bottlenecks of the road network, where multiple traffic streams and people flow gather [2]. Traffic congestion usually occurs at intersections. There are many causes of congestion, and the primary cause is supply and demand imbalance. Engineers often improve the

intersection capacity through refined design and signal timing optimization. The capacity theory can help engineers make the right decisions about design, planning and operation [3]. Therefore, capacity analysis plays an important role in engineering applications.

Saturation flow rate (SFR) is one of the most critical parameters in the capacity estimation field at signalized intersections. The U.S. Highway Capacity Manual (HCM) defines the saturation flow rate. It represents that the maximum flow rate in lane groups per hour when the traffic signal will always display green [3]. The saturation flow rate cannot be measured directly and needs to be estimated by traffic engineers. Although different intersections have different saturation flow rates, the same intersection also has different saturation flow rates. It is because the saturation flow rate is affected by many factors such as geometric conditions, traffic composition and pedestrian interference. It becomes necessary for estimating the saturation flow rate. At present, two methods for calculating the saturated flow rate are proposed in U.S. HCM. One is the adjustment method. The other is field measurement method. The adjustment method is a multiplication formula, which contains base saturation flow rate (BSFR) and adjustment factors. The base saturation flow rate refers to the saturation flow rate under the ideal conditions for each influencing factor. The adjustment factors refer to the degree of influence of various factors on saturation flow rate. The filed measurement method is to obtain saturation flow rate by calculating the time reciprocal of saturation headway. The saturation headway is the average headway which is the mean value from 4–6th vehicle to last vehicle headway in the queue.

At present, researchers have proposed many estimated saturation flow rate models, which are mainly divided into the following three categories. One is a type of model based on the adjustment method in HCM [4–8] (see Section 2). In consideration of the traffic characteristics of different areas, they revise the existing adjusted factors and add new adjusted factors to improve the original adjustment model. This type of model can clearly express the relationship between influencing factors and saturation flow rates and well assist engineers to make the right decisions. The second one is a type of model based on the estimated departure headway method [9–12] (see Section 2). The key to this type of model is to determine whether the vehicle's headways in the queue are saturated. Then, the saturated headways are converted into saturation flow rates. Normally, the saturated headway is related to vehicle size and queue position. This type of model can estimate each position vehicle headway. The last one is a type of model based on statistical and physical methods [13–16] (see Section 2). They show the relationship between saturation flow rate and influencing factors by a new perspective. However, the above models fail to consider different traffic conditions and influencing factors for real-time prediction in the actual traffic network.

Nevertheless, the previous research findings provide an insightful approach for the study of saturation flow rate estimation at a signalized intersection. In response to the above needs, the research efforts in this paper have been dedicated to the following aspects:

(i) Developing a better model to depict the relationship between saturation flow rate and influencing factors in different traffic conditions.

(ii) Discussing the different traffic characteristics and proposing an effective method to estimate the saturation flow rate dynamically.

The remainder of this paper is organized as follows. Section 2 provides an overview of the existing solutions for saturation flow rate estimation models. In Section 3, we provide our proposed neural network. Section 4 presents the collection data observed in Beijing. Section 5 verifies this new probability model based on collection data and compares it with the conventional method. Finally, we conclude this paper in Section 6.

## 2. Related Works

### 2.1. U.S. Highway Capacity Manual Methods

In the world, the capacity analysis method of signalized intersections at different country are based on U.S. Highway Capacity Manual methods. HCM has been updated five times so far. In HCM,

the saturation flow rate of signalized intersection is the sum of the saturation flow rate for each lane group of each approach. Two conventional methods for determining the saturation flow rate are proposed in the U.S. Highway Capacity Manual [3]. One is the adjustment method. The other is measurement technique. In the adjustment method, the computed saturation flow rate is referred to as the "adjusted" saturation flow rate because it reflects the application of various factors that adjust the base saturation flow rate to the specific conditions present on the subject intersection approach. Equation (1) is used to compute the adjusted saturation flow rate per lane for the subject lane group.

$$s = s_0 f_w f_{HV} f_g f_p f_{bb} f_a f_{LU} f_{LT} f_{RT} f_{Lpb} f_{Rpb}, \tag{1}$$

where $s_0$ is the base saturation flow rate (pc/h/ln); $f_w$ is the adjustment factor for lane width; $f_{HV}$ is the adjustment factor for heavy vehicles in traffic stream; $f_g$ is the adjustment factor for the approach grade; $f_p$ is the adjustment factor for the existence of a parking lane and parking activity adjacent to lane group; $f_{bb}$ is the adjustment factor for blocking effect of local buses that stop within the intersection area; $f_a$ is the adjustment factor for area type; $f_{LU}$ is the adjustment factor for lane utilization; $f_{LT}$ is the adjustment factor for left-turn vehicle presence in a lane group; $f_{RT}$ is the adjustment factor for right-turn vehicle presence in a lane group; $f_{Lpb}$ is the pedestrian adjustment factor for left-turn groups; $f_{Rpb}$ is the pedestrian-bicycle adjustment factor for right-turn groups. In general, the adjustment method is used to estimate the saturation flow rate at planning intersections.

In the measurement technique, data is taken cycle by cycle. In general, vehicles are recorded when their front axles cross the stop line. Saturation flow rate is calculated only from the data recorded after the fourth to sixth vehicle in the queue passes the stop line. To reduce the data for each cycle, the time recorded for the fourth vehicle is subtracted from the time recorded for the last vehicle in the queue. This value represents the sum of the headways for the fifth through $n$-th vehicle, where n is the number of the last vehicle surveyed (which may not be the last vehicle in the queue). This sum is divided by the number of headways after the fourth vehicle (i.e., divided by $(n-4)$) to obtain the average headway per vehicle under saturation flow. The saturation flow rate is 3600 divided by this average headway as expressed in Equation (2). In general, the measurement technique is used to obtain a saturation flow rate at operating intersections.

$$s = \frac{3600}{h_t}, \tag{2}$$

where, $h_t$ is saturation headways.

## 2.2. Improved Adjustment Methods

In order to improve the applicability of HCM adjustment locally, researchers have focused on adjustment factors. In China, owing to limitations on land use, many intersections are irregular crossing where the approach and exit lanes are offset or two roads cross at oblique angles. So the guideline markings are set in signalized intersections. However, this special factor is not taken into account to modify the base saturation flow rate in HCM. Qin et al [4] proposed a new adjustment factor for guideline markings. The saturation flow rate could be estimated accurately with intersection guideline markings. It was found that painting guideline markings could improve the saturation flow rate at signalized intersections. In most of the developing countries like India, two-wheelers are the major mode of personal transportation. Therefore, their effect on the saturation flow at signalized intersections could be substantial. It is not possible to use the U.S. Highway Capacity Manual model directly. Anusha et al [5] proposed a new adjustment factor for two-wheelers and incorporated the effect of two-wheelers on the saturation flow rate into a previous model. It was found that the estimation of the saturation flow rate using the modified model was closer to observed values. Due to the difference in roadway conditions, the driver's behavior and cultures etc, the parameter values of HCM are not applicable in suitable in China. Shao et al. [6] collected 18 cities' data. Then, base saturation flow rate,

adjustment factors for lane width and approach grade were suggested. The adjustment factor for turn radius and PCE (passenger car equivalents) were developed. Wang et al. [7] and Lewis et al. [8] found that there was an interaction effect between heavy vehicles and lane width. Wang et al. [7] proposed a comprehensive adjustment factor for considering the interaction. It effectively improved the estimation accuracy.

The above methods maintain well the structure of the HCM model. They can directly express the influencing factors and their effect level on the saturation flow at signalized intersections. As such, the relationship between each factor and the saturation flow rate needs to be surveyed. With respect to geometric factors, the intersection is usually not rebuilt. The engineers can collect data once in a period. With respect to traffic factors, the traffic composition, driver's behavior, pedestrians and bicycle interference are different in each cycle. Therefore, a large number of engineers are required for data collection resulting in a high economic cost.

## 2.3. Estimated Departure Headway Methods

In general, departure headways at signalized intersections are defined as the time intervals between successive vehicles in a queue passing a stop line or a reference line at the intersection. Many researchers have focused on the recognition of saturation flow and achieved automatic extraction. Yang et al. [9] proposed a dynamic extraction method of saturation headway by using an induction coil detector. The average saturation headways of history and current cycle were analyzed with an exponential smoothing method. Compared with the traditional methods, the proposed method can be realized without additional cost and can meet the demand of dynamic extraction. Wang et al. [10] proposed an automatic estimation method for the saturation headway based on video detector data. The Dickey–Fuller test was used to verify whether the headways in the time series were saturation headways. An iterative method using quantiles was proposed to filter out abnormal data. The quantile of 80% and the data duration of no less than 150 min were suggested. In addition, Radhakrishana et al. [11] analyzed the factors affecting discharge headway under heterogeneous traffic conditions and proposed a novel method to measure headways. It was found that discharge headway values were having variation and were different from the homogeneous traffic scenario where the headway tends to follow a constant value after the initial four or five vehicles. Models for computing discharge headway were developed using linear regression and linear mixed regression. Differently, Tong et al. [12] proposed a neural network approach to estimate the queued vehicle discharge headway.

The above methods are based on the historical or current cycle of data extraction. They are difficult to predict the saturation headways in the next cycle. Due to the limitation of data collection equipment, many influencing factors are difficult to consider in real-time. In recent years, video detectors have been widely placed nearby the signalized intersections. The video detectors can automatically record vehicle crossing stop line time, vehicle speed and other information (like pedestrians and bicycles). The roadside units (RSU), which were placed in the intersections, can also record the vehicle type, location, trajectory and other information [17–19]. In the connected vehicle environment, the status information of each vehicle can also be collected [20]. Two types of data are mutually complemental. We can obtain the queuing situation and traffic composition from connected vehicle data. In an information-rich environment, automatic estimation of saturation flow rate at all intersections in cities can be achieved without a manual survey resulting in low economic costs. In China, the government promotes the electronic toll collection (ETC) equipment installed in all vehicles [21]. At present, the number of ETC users has exceeded 180 million. In Chongqing, all vehicles need to be set up radio frequency identification device (RFID) license plates [22]. At the same time, with the development of new transportation modes like mobility as a service (MaaS), the vehicle information will be more open. According to Accenture's report [23], in 2030, the format of the automotive industry will change from marketing to transportation services. These developments of new technologies will support the prediction of saturation headways.

### 2.4. Statistical and Physical Methods

Some researchers analyze the saturation flow rate form a new perspective, and develop statistical models of influencing factors and saturation flow rate, and establish physical models of microscopic traffic characteristics. Saha et al. [13] proposed a new saturation flow model that can accurately estimate the saturation flow rate in India traffic conditions. This model presents four forms—universal Kriging, pseudo-likelihood Kriging, blind Kriging, co-Kriging based saturation flow model. The performance of the new model is better than the conventional model. Shao et al. [14] studied the randomness of the saturation flow rate with a statistical method. It was found that the result of calculating the saturation flow rate may be low with the average value of saturation headways. A new method of estimation saturation flow rates is developed based on the median value of queue discharge headways. Murat et al. [15] built a new model for saturation flow based on driver behavior and vehicle characteristics. The vehicle's length, acceleration of vehicles, speed of vehicles crossing intersection and reaction time of vehicles in the queue are considered. There is no significant difference between the new model and measured data. Reaction time is the major factor in saturation flow. Vehicle length is also an important factor. Hossain [16] developed a regression model to estimate the saturation flow rate. It was found that the saturation flow rates were dependent on the lane width, percentage of turning vehicles, and the composition of vehicles. Chen et al [24] proposed a four-stage saturation departure model based on an empirical analysis of discharge headways in shared lanes. The model reflects the stochastic nature of vehicle-pedestrian conflict and constructs the logical relationships between shared right-turn saturation flow and its influencing factors.

The above models are developed under specific traffic conditions. They can describe the relationship between saturation flow and influencing factors. However, these models are relatively complex and fail to be used in engineering applications. Among these models, the performance of the regression models are acceptable in simple traffic scenarios with few influencing factors. However, in the complex traffic scenarios, the regression models are difficult to fit and the accuracy of the estimation of the saturation flow rate will decrease significantly.

In summary, previous studies provided an approach for estimating saturation flow from different perspectives. However, the previous research models usually used historical data in a period to estimate the saturation flow rate. It did not predict the saturation flow cycle by cycle. In addition, under complex traffic scenarios, the accuracy of the traditional method for estimating saturation flow rates are unsatisfactory. In this study, we combined the advantages of the adjustment method and field measurement method and proposed a neural network model that can predict the saturation flow rate cycle by cycle. The results of this study can improve the accuracy of estimation of saturation flow rate, dynamically optimize signal control schemes, and improve the level of traffic management.

## 3. Methods

### 3.1. Conventional Method

In this paper, we use two traditional methods to estimate the saturation flow rate for comparison of the proposed neural network method. They are the adjustment model and the multiple linear regression model. With respect to the adjustment model, each adjustment factor expressed the relationship between the influencing factor and saturation flow rate. The adjustment factors were divided into dynamic factors and static factors. The static factors are geometric factors such as lane width. There are differences between different intersections. The dynamic is traffic factors such as heavy vehicles in the traffic stream. The adjustment factors in the HCM are mainly applicable to American traffic conditions. In China, a Chinese National Standard (code for the planning of intersections on urban roads, Standard Number is GB50647-2011) [25] was published. It was based on HCM. It took into account the characteristics of Chinese transportation. Equations (3)–(5) are used to compute the through lane, left-turn lane, right-turn lane saturation flow rates in China, respectively.

$$S_t = S_{bt} \times f_t \times f_g \tag{3}$$

$$S_l = S_{bl} \times Min[f_t, f_z] \times f_g \tag{4}$$

$$S_r = S_{br} \times Min[f_t, f_z] \times f_g, \tag{5}$$

where, $S_t$ is the adjusted saturation flow rate in through lane; $S_l$ is the adjusted saturation flow rate in the left-turn lane; $S_r$ is the adjusted saturation flow rate in the right-turn lane; $S_{bt}$ is the base saturation flow rate in through lane; $S_{bl}$ is the base saturation flow rate in the left-turn lane; $S_{br}$ is the base saturation flow rate in the right-turn lane; $f_t$ is the adjustment factor for lane width; $f_g$ is the adjustment factor for approach grade and heavy vehicles in the traffic stream; $f_z$ is the adjustment factor for turning radius.

### 3.2. Neural Network Method

Since 1990, neural networks have been widely used in transportation [26]. Neural networks have two advantages that are suitable for describing complex traffic flow. One advantage is that the neural network can handle any complexity nonlinear systems. The other is that it does not require any prior knowledge for model building. For example, in regression models, basic relationships (linear, polynomial, exponential, etc.) need to be determined before model building. Therefore, neural networks are more flexible than statistical methods. The basic elements of the neural network are nodes which are arranged in a layered structure. Nodes are connected to each other through links. Each link is associated with a weight. Each node may receive many inputs from other nodes, applying an activation function to its input to determine its output signal. The sigmoid function (see Equation (6)), which is one of the typical activation functions, was chosen in this study.

$$f(a) = \frac{1}{1 + e^{-a}}. \tag{6}$$

A typical fully connected three-layered neural net is shown in Figure 1. The first layer is the input layer that contains the input units $x_i$ (n nodes). The inputs are linked to the nodes in the hidden layer $z_j$ (p nodes) with associated weight $\omega_{ij}$. The nodes in the hidden layer are also linked to the output nodes $y_k$ with associated weight $v_{jk}$. The output units and hidden units may also have bias units. The $b_j$ and $bb_k$ are the bias of the hidden layer and output layer, respectively. The neural net can be trained through the adjustment of weights by input training patterns.

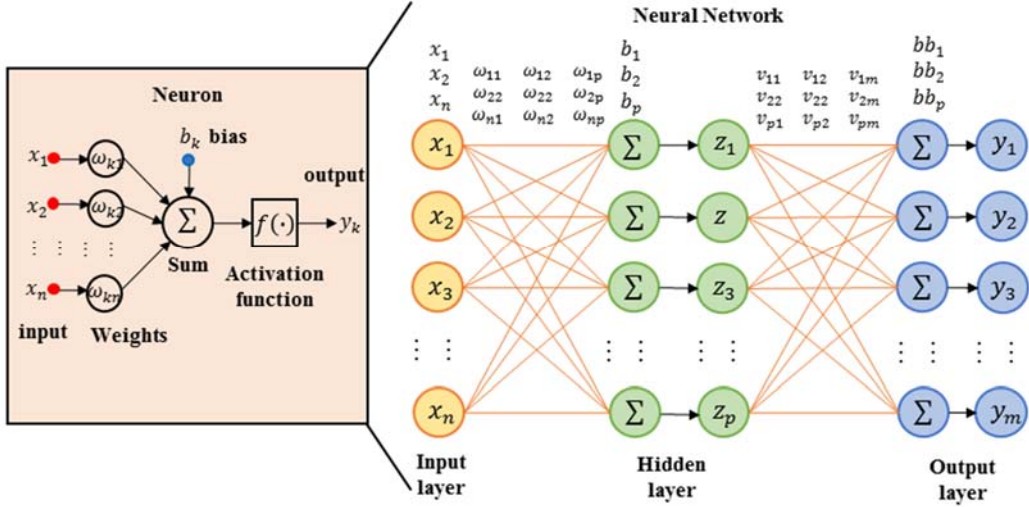

**Figure 1.** Structure of neural network.

### 3.2.1. Training of the Neural Network

Currently, there are many available training algorithms, and the backpropagation rule is selected in this study, which is one of the most widely used training algorithms to deal with prediction problems. The principle of this rule is to minimize the total output error with respect to the weight. The network adjusts the weights in the direction that reduces the error. The training rules are as follows [12]:

Step 1: Select a learning rate $\alpha$ and initialize the weight $\omega_{ij}$ and $v_{jk}$ using random values.

Step 2: Present the input data and compute the layers' output by Equations (7) and (8).

$$z_i = f\left(\sum_{i=1}^{n} \omega_{ij} x_i + b_j\right) \tag{7}$$

$$y_k = f\left(\sum_{j=1}^{p} v_{jk} z_j + bb_k\right) = f\left(\sum_{j=1}^{p} v_{jk}\left(\sum_{i=1}^{n} \omega_{ij} x_i + b_j\right) + bb_k\right). \tag{8}$$

Step 3: Compute the weight correction terms of output and hidden units by Equations (9) and (10).

$$\Delta\omega_{ij} = \alpha f'\left(\sum_{i=1}^{n} \omega_{ij} x_i + b_j\right)\sum_{k=1}^{m}\left\{(t_k - y_k) f'(\sum_{j=1}^{p} v_{jk} z_j + bb_k) v_{jk}\right\} x_{ij} \tag{9}$$

$$\Delta v_{jk} = \alpha(t_k - y_k) f'\left(\sum_{j=1}^{p} v_{jk} z_i + bb_k\right) z_j. \tag{10}$$

Step 4: If all the input data patterns are trained, go to step 5, otherwise, go to step 2.

Step 5: Sum all the weight correction terms and then add correspondingly to the old weights.'

Step 6: Repeat steps 2–5 until a sufficiently small output error has been obtained.

Each iteration from steps 2–6 is called an epoch. Usually, many epochs are required between training is completed. The hidden layer nodes, activation function, learning rate, and gradient descent function are effect on speed and accuracy of the training process. The number of nodes for the output layer is one. It is measured the saturation flow rate. The number of nodes for the hidden layer is based on Equation (11).

$$n_1 = \sqrt{n + m} + a, \tag{11}$$

where, $n$ is the number of input layer nodes; $m$ is the number of output layer nodes; $a$ is a constant (range from 1 to 10). This paper determines the range of hidden layer neuron nodes according to Equation (11).

### 3.2.2. Performance Evaluation

In order to validate the trained neural network, the data set had to be divide into two subsets, namely the training and test sets. The training set was used to train to network and check whether the network is over-fitted or not. Several types of prediction errors were available to assess the performance of the neural network such as the mean squared error and mean absolute percentage error [12]. In the present study, the mean absolute percentage error (MAPE) was used to evaluate the model performance, which is shown in Equation (12).

$$MAPE = \frac{1}{n}\left(\sum_{i=1}^{n} \frac{|E_i - M_i|}{M_i}\right) \cdot 100\%, \tag{12}$$

where $E_i$ is the model output value, $M_i$ is the measured value, $n$ is the number of testing data set.

### 3.3. Proposed Model

The modeling technique proposed in this research is based on the main factors affecting the saturation flow rate. To estimate the saturation flow rate of each cycle, a three-layered neural network with one output unit was used. The output unit was the measured saturation flow rate.

#### 3.3.1. Selection of Input Variables

The influencing factors of the saturation flow rate are different in different traffic scenarios. As is shown in Figure 2, the intersection users compete for conflict space, further explaining the characteristics of each factor. In through lanes, the saturation flow rate is affected by few factors. It is the composition of vehicles in the queue. However, it was found that there was the mutual interference among vehicles in multiple through lanes. After passing the stop line, the vehicle freely chose the lanes of departure. It affected the running vehicles and caused fluctuation of the saturation flow rate. In shared right-turn and through lanes, both through and right-turn vehicles can also cross the stop line. Therefore, the saturation flow rate was not only affected by the composition of vehicles, but also by the percentage of right-turning vehicles. When the right-turning vehicles crossed the departure, it was also disturbed by pedestrians and bicycles. The influencing factors in shared left-turn and through lanes were similar to the shared right-turn and through lanes. They were the percentage of left-turning vehicles, disturbing of pedestrians and bicycles, respectively. However, saturation flow rate was also affected by opposing vehicles.

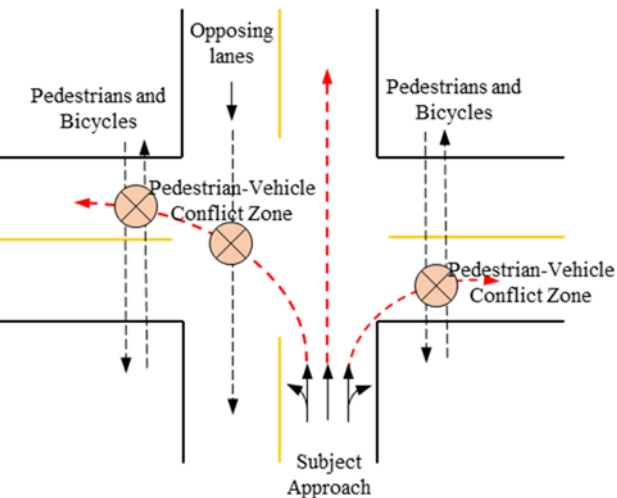

**Figure 2.** Conflict zone locations.

In general, the influencing factors are used as input variables. For variables that have been selected, it must be easy to measure on-site since a considerable amount of data has to be collected to develop the model. Based on this objective, seven variables were chosen to be input variables for the estimation of the saturation flow rate in each cycle. They are shown in Table 1. We should choose different variables in different scenarios.

**Table 1.** Definition of input variables.

| Variables | Definition | Ranges |
|---|---|---|
| $X_1$ | Lane width (m) | No specific range |
| $X_2$ | Percentage of heavy vehicles | 0–0.5 |
| $X_3$ | Interference in multiple through lanes | 0, 1 |
| $X_4$ | Percentage of turning vehicles in the lane group | 0–1.0 |
| $X_5$ | Disturbed pedestrians | No specific range |
| $X_6$ | Disturbed bicycles | No specific range |
| $X_7$ | Opposing vehicles | No specific range |

### 3.3.2. Model Structure

The structure and information of the model are shown in Figure 3.

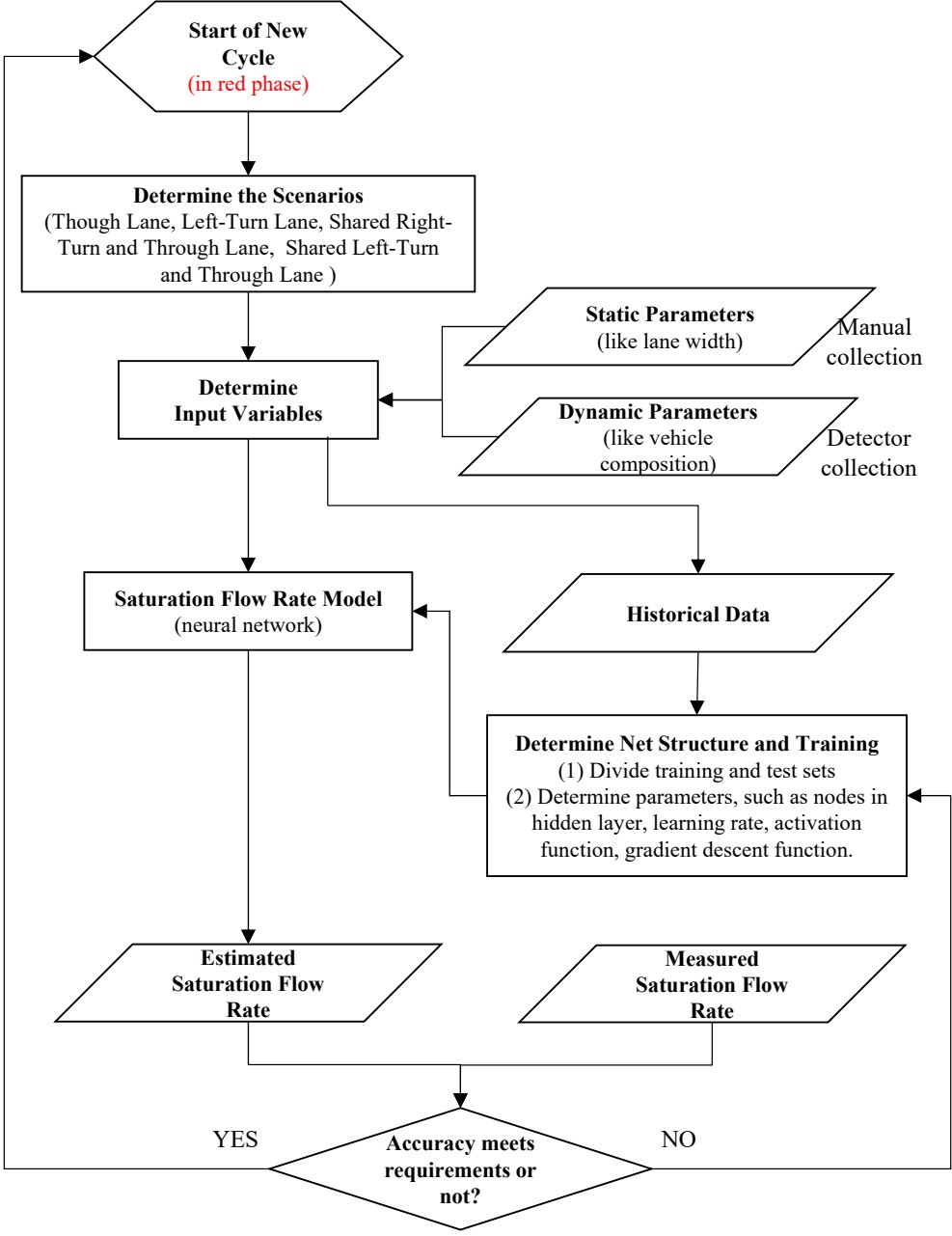

**Figure 3.** The structure and information of proposed model.

Firstly, it is necessary to obtain data. Since the saturation flow rate is a key parameter of signal timing, it is very important to obtain the saturation flow rate in real-time. The characteristics of the saturated flow rate and traffic volume were different. The traffic volume was measured in real-time at 5 mim or 15 min intervals. Because the saturation flow rate was related to the signal period, it is more appropriate to use each signal period as the statistical interval. According to the characteristics of the neural network model, the input and output parameters need to be determined. In this paper, the input parameters are the influencing factors, and the output parameters are the measured saturation flow rates. The output parameters can be obtained through detectors (video camera, roadside unit etc.) nearby the intersections. There were four steps in the measurement method. (a) We should determine whether the number of queued vehicles exceed seven. If it is less than seven, it will indicate that the headway is not reached saturation. We could not obtain the saturation headway directly. (b) We should determine whether the headway, which ranged from the fourth vehicle to the last vehicle in the queue, was saturation headway. (c) The average saturation headway can be extracted. (d) The saturation flow rate was calculated by Equation (2). There were two types of input parameters, one a static parameter and the other is a dynamic parameter, which were obtained through different methods. Static parameters can be collected manually and updated every year. Dynamic parameters can be collected by detectors.

Secondly, the structure of the neural network is determined. The number of nodes for the input layer was the same as all influencing factors (static and dynamic factors). The number of nodes for the output layer was one. The saturation flow rate was measured. Then, the optimal number of neuron nodes in the hidden layer was determined by judging the minimum error. Similarly, the activation function, gradient descent function and learning rate were determined according to the experiment. Finally, a neural network model can be built by training and verifying. The training may spend a lot of time. We could retrain and optimize the neural network at some intervals. The neural network model for dynamic estimation of the saturation flow rate was divided into five steps in this paper. As is shown in Figure 4.

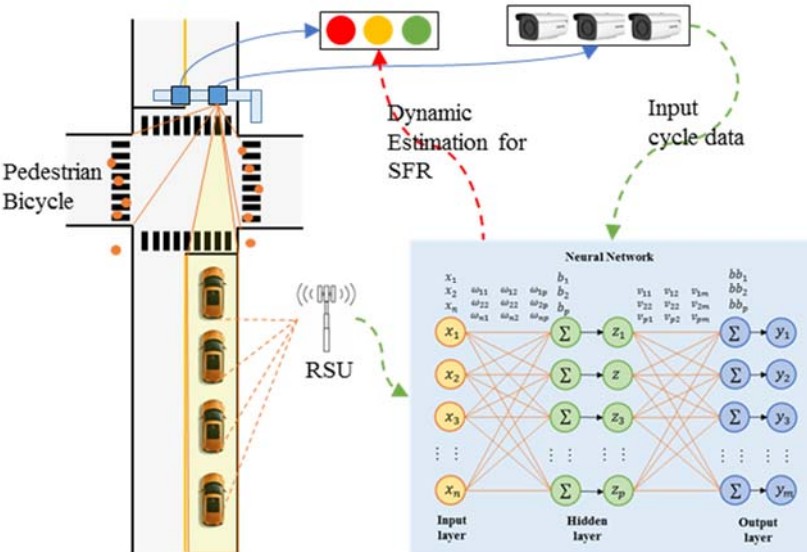

**Figure 4.** Application of neural network model at signalized intersection.

Step 1: According to the traffic situation, we should analyze the influencing factors of subject lanes and determine input parameters.

Step 2: According to the parameters of step 1, the structure of a neural network, such as neural nodes in a hidden layer, activation function, gradient descent function, learning rate, can be determined.

Step 3: We use one part of historical data as a training set. After training, optimizing, iterating to convergence, the neural network model is built. We use another part of historical data as a verifying set. After verification, the parameters, weight matrix and bias matrix are saved.

Step 4: We use the trained neural network model and measured data of the current signal cycle to predict the saturation flow rate in the next cycle.

Step 5: We determine whether the neural network can update. If it is satisfied, it would go to step 3, otherwise, it would go to step 4.

## 4. Data Collection

In order to prove the availability and reliability of the proposed model in this paper, different traffic scenarios are selected. Then, the neural network model, HCM model, and regression models for estimating saturation flow rate were developed based on measured data. According to the characteristics of traffic flow, the influencing factors were different in various lanes. Therefore, the traffic flow characteristics and the number of influencing factors should be considered when selecting the traffic scenarios. Finally, three scenarios of the approach are through lane, shared left-turn and through lane, shared right-turn and through lane. The scenario 1 (through lane) is located at the intersection of Shiliuzhuang road and Liuxiang Road, Fengtai district, Beijing. There was heavy traffic volume during peak hours at this intersection. Scenario 2 (shared right-turn and through lane) is located at the intersection of Chegongzhuangxi road and Shoudutiyuguannan road, Haidian district, Beijing. There are many pedestrians and bicycles at this intersection. Scenario 3 (shared left-turn and through lane) is located at the intersection of Andingmenwai road and Waiguanxie street, Dongcheng district, Beijing. It is surrounded by a residential area, with a large pedestrian flow.

The three intersections are shown in Figure 5. The detailed geometric features and signal timing schemes of approach at each intersection are shown in Table 2. In the scenario 1 intersection, there are four through lanes, one right-turn lane and one left-turn lane in the northbound approach. There is one U turn lane, one left-turn lane, five through lanes and one right-turn lane in southbound approach. During peak hours, the signal timing scheme was four phases. Right-turn vehicles were not controlled by the signal light. Left-turning vehicles were separated from through vehicles. The signal cycle time is 140 seconds. The amber light time was 4 and 3 s in through phase and left-turn phase, respectively. The all-red interval is 2 s. The green light time was 44 s (northbound and southbound through phase), 42 s (eastbound and westbound through a phase), 15 s (northbound and southbound left-turning phase) and 17 s (eastbound and westbound left-turning phase), respectively. Nine through lanes were selected at the scenario 1 intersection in this paper. In the scenario 2 intersection, there were two through lanes, one left-turn lane, one right-turn lane and one shared right-turn and through lane in the westbound approach. During peak hours, the signal timing scheme was four phases. Right-turn vehicles were not controlled by signal light. Left-turning vehicles were separated from through vehicles. The signal cycle time was 148 s. The amber light time was 3 s in all phases. The all-red interval was 2 s. The green light time was 40 s (eastbound and westbound through phase) and 20 s (eastbound and westbound left-turning phase), respectively. One shared right-turn and through lane is selected at the scenario 2 intersection in this paper. In the scenario 3 intersection, there is one shared left-turn and through lane and one right-turn lane in the eastbound approach. During peak hours, the signal timing scheme was three phases. The signal cycle time was 156 s. The amber light time was 3 s in all phase. The all-red interval was 2 s. The green light time was 42 s (eastbound and westbound through phase, left-turn phase). One shared left-turn and through lane was selected at the scenario 3 intersection in this paper.

**Table 2.** Location of three traffic scenarios and the detailed characteristics.

| | Scenario 1:<br>TH [1] | Scenario 2:<br>TH-RT [2] | Scenario 3:<br>TH-LT [3] |
|---|---|---|---|
| Intersection Name | Shiliuzhuang Rd [4] and Liuxiang Rd | Chegongzhuangxi Rd and Shoudutiyuguannan Rd | Andingmenwai St [5] and Waiguanxie St |
| Westbound Lanes in Approaches | 1U [6]+2LT [7]+2TH+1RT [8] | 1LT+2TH+1TH-RT+1RT | 1LT-TH-RT |
| Eastbound Lanes in Approaches | 1LT+1TH+1RT | 1LT+2TH+1RT | 1LT-TH+1RT |
| Northbound Lanes in Approaches | 2LT+4TH+1RT | 1LT+3TH+1RT | 1LT+3TH+1B [9]+1TH-RT |
| Southbound Lanes in Approaches | 1U+1LT+5TH+1RT | 2LT+2TH+1LT+1RT | 1LT+3TH+1B+1TH-RT |
| Cycle Time | 140s [10] | 148s | 156s |
| Phase Number | 4 | 4 | 3 |
| Eastbound and Westbound Through Phase | 42s (green) + 4s (amber) + 2s(all-red) | 40s (green) + 3s (amber) + 2s(all-red) | 42s (green) + 3s (amber) + 2s(all-red) |
| Eastbound and Westbound Left-turn Phase | 17s (green) + 3s (amber) + 2s(all-red) | 20s (green) + 3s (amber) + 2s(all-red) | |
| Northbound and Southbound Through Phase | 44s (green) + 4s (amber) + 2s(all-red) | 48s (green) + 3s (amber) + 2s(all-red) | 66s (green) + 3s (amber) + 2s(all-red) |
| Northbound and Southbound Left-turn Phase | 15s (green) + 3s (amber) + 2s(all-red) | 20s (green) + 3s (amber) + 2s(all-red) | 33s (green) + 3s (amber) + 2s(all-red) |

[1] TH express through lane. [2] TH-RT express shared right-turn and through lane. [3] TH-LT express shared left-turn and through lane. [4] Rd express road. [5] St express street. [6] U express U turn lane. [7] LT express exclusive left-turn lane. [8] RT express exclusive right-turn lane. [9] B express bus transit lane. [10] s express second.

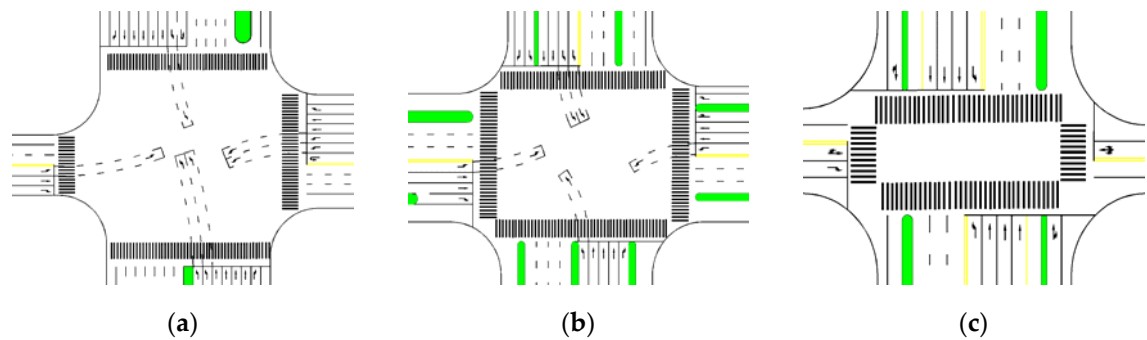

**Figure 5.** Channelization of signalized intersections: (**a**) scenario 1 intersection; (**b**) scenario 2 intersection; (**c**) scenario 3 intersection.

In view of so much data, a video camera was used to record the vehicle's movements at the intersections in this paper. The vehicle's headways and influencing factors data were extracted manually. The lane width was recorded by a range finder. The signal cycle time and phases were recorded by a stopwatch. In the extraction process, the influencing factors and headways were recorded each cycle. If the vehicles in the subject lane affected by an adjacent vehicle's lane changes, we recorded 1, otherwise 0. A total of 420 cycle data were collected for the nine through lanes in the evening peak (17: 30–19: 30) from 2 to 4 April 2019. A total of 90 cycle data were collected for the shared left-turn and through lane and shared right-turn and through lane in the evening peak (17: 30-19: 30) from 12 to 14 June 2019, respectively.

## 5. Results and Discussion

### 5.1. Data Summary

A total of 600 cycle data were collected in this paper. The details about the data are shown in Table 3. There were 420 cycle data which were saturation headways, percentage of heavy vehicles and adjacent vehicle's lane changes in scenario 1. The average saturation flow rate was 1399 veh/h. The maximum saturation flow rate is 2030 veh/h. The minimum saturation flow rate is 967 veh/h. The standard deviation is 195.09. There were 90 cycles data which were saturation headways, percentage of heavy vehicles, percentage of right-turning vehicles and pedestrians and bicycle volumes in scenario 2. The average saturation flow rate was 1355 veh/h. The maximum saturation flow rate was 2105 veh/h. The minimum saturation flow rate was 701 veh/h. The standard deviation was 292.67. There were 90 cycles data which were saturation headways, percentage of heavy vehicles, percentage of left-turning vehicles and pedestrians and bicycle volumes in scenario 3. The average saturation flow rate was 1395 veh/h. The maximum saturation flow rate was 1782 veh/h. The minimum saturation flow rate was 960 veh/h. The standard deviation is 177.17.

**Table 3.** Descriptive statistics of field saturation flow rates.

| Summary | Scenario 1 | Scenario 2 | Scenario 3 |
|---|---|---|---|
| Count | 420 | 90 | 90 |
| Average Saturation Flow Rate (veh/h) | 1398.77 | 1335.91 | 1395.15 |
| Maximum Saturation Flow Rate (veh/h) | 2030.62 | 2105.26 | 1782.18 |
| Minimum Saturation Flow Rate (veh/h) | 967.74 | 701.75 | 960.00 |
| Standard Deviation | 195.09 | 292.67 | 177.17 |

### 5.2. Saturation Flow Rate Estimation Model with a Neurnal Network

A three-layer neural network model was proposed. According to the measured data, with the goal of minimizing the error of the output result, the parameters of the input layer, hidden layer and

output layer are calibrated. The number of hidden layer nodes, activation function, gradient descent function, and learning rate is selected. The neural network model program was developed by the TensorFlow which is an open-source software by Google. The software had the standardized neural network architecture and functions for coding conveniently. At the same time, the calculation process and results were visualized through TensorBoard. It could make the optimization and debugging of the program easy. Its interface was shown in https://tensorflow.google.cn/.

Firstly, the input and output variables were determined. In general, the influencing factors were used as the input variables, and the measured saturated flow rates were used as the output variables. The input variables were different in different scenarios, and the output variables were the measured saturated flow rates. Scenario 1: The three input variables were lane width, vehicle composition, and multi-lane lateral interference. Scenario 2: The four input variables were vehicle composition, percentage of right-turning vehicles, pedestrians, and bicycles. Scenario 3: The four input variables were the percentage of left-turning vehicles, pedestrians, bicycles and opposing vehicles. Secondly, the training and test set were divided. 300 cycles data were used to train model and 120 cycles data were used to test model in scenario 1. 65 cycles data were used to train model and 25 cycles data were used to test model in scenarios 2 and 3. Then, the hyperparameters in the neural network model were calibrated and saved. In TensorFlow, the adjusted hyperparameters included the number of hidden layers, the number of neural nodes, the activation function, the learning rate, and the gradient descent function. In general, the number of hidden layers was one. The calibration process of hyperparameters is as follows.

Step 1: The range of hyperparameters were determined. For example, the number of neural nodes could be calculated according to Equation (9). The range was from 3 to 12. In general, there are 6 types of activation functions. The learning rate ranged from 0.01 to 0.1. There are 6 gradient descent functions.

Step 2: We should control the hyperparameters for model training and testing. We could choose the number ranging from 3 to 12 in turn to train. The structure and parameters were saved. Then, we used the same structure and parameters to test the model. The predictions of mean absolute error (MAE) and mean absolute percentage error (MAPE) were calculated.

Step 3: The prediction indexes and hyperparameters were compared and determined. For different scenarios, the prediction indexes were compared in the test set, and the hyperparameters were selected in which the prediction indexes were smallest.

Taking the model developed in scenario 1 as an example, the hyperparameters were determined by the above steps as shown in Figure 6. The number of neural nodes was 12. The activation function was "sigmoid". The learning rate was 0.04. The gradient descent function was "RMSProOptimizer". Scenarios 2 and 3 were similar to scenario 1. The results were shown in Table 4. In fact, the hyperparameters were determined automatically by programming.

After all hyperparameters were determined, the models of three scenarios were trained with training sets. After training, the model weights and bias matrix were saved. Then, we used the same parameters to verify the model with test sets. In scenario 1, the weight matrix was expressed as "$W_{11}$" and the bias matrix was expressed as "$b_{11}$" between the input layer and the hidden layer. The weight matrix was expressed as "$W_{12}$" and the bias matrix was expressed as "$b_{12}$" between the hidden layer and the output layer. Similarly, the scenario 2 matrixes were expressed as "$W_{21}, b_{21}, W_{22}, b_{22}$". The scenario three matrixes were expressed as "$W_{31}, b_{31}, W_{32}, b_{32}$". They are shown in Appendix A.

The prediction results of training and test sets in three scenarios were shown in Figure 7. The predicted and measured saturation flow rates in training and test sets were in good agreement. In scenario 1, the mean absolute percentage error values were 0.06 (94% accuracy) and 0.11 (89% accuracy) with training and test sets, respectively. In scenario 2, the mean absolute percentage error values were 0.07 (93% accuracy) and 0.07 (93% accuracy) with training and test sets, respectively. In scenario 3, the mean absolute percentage error values were 0.04 (96% accuracy) and 0.05 (95% accuracy) with training and test sets, respectively.

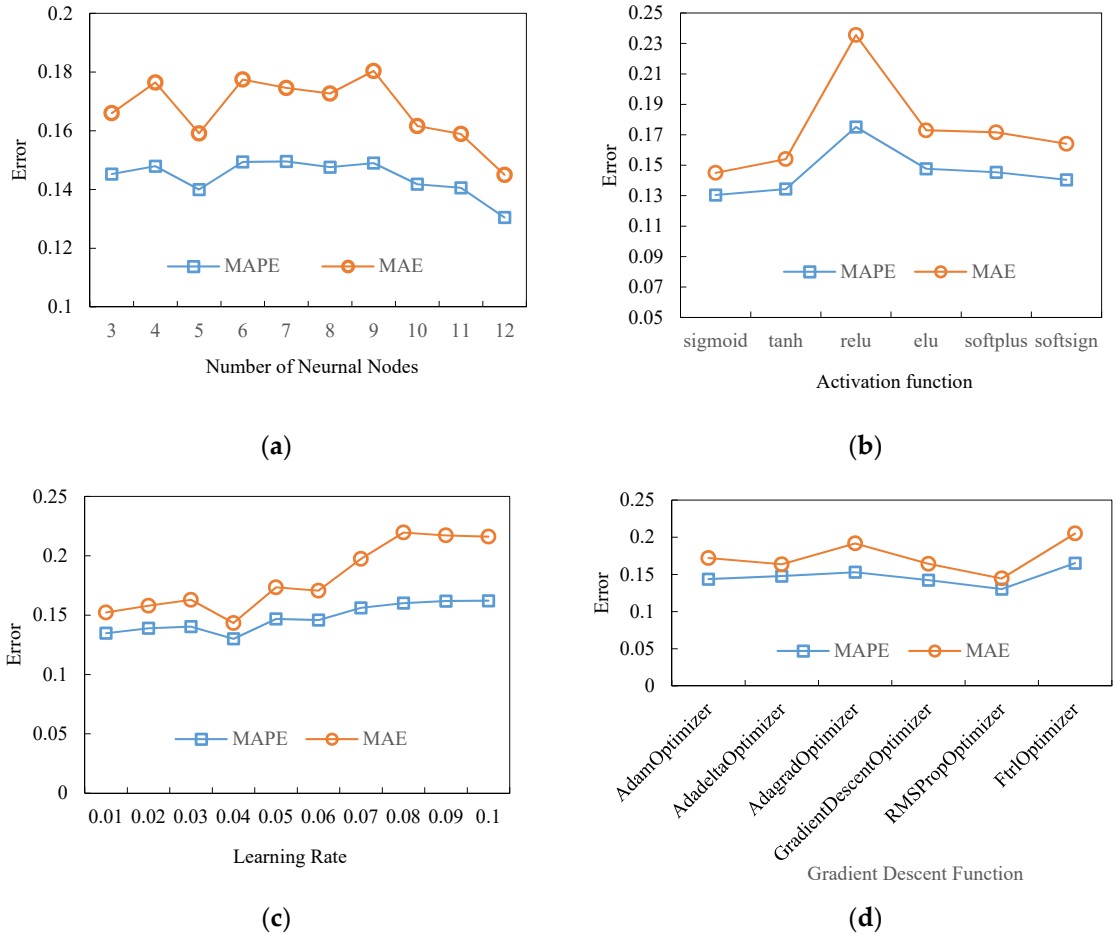

**Figure 6.** Determining the hyperparameters of neural network models in scenario 1: (**a**) error changes of number of neural nodes; (**b**) error changes of activation function; (**c**) error changes of learning rate; (**d**) error changes of gradient descent function.

**Table 4.** Choices of hyperparameters for neural network models at three scenarios.

|  | Scenario 1 Model | Scenario 2 Model | Scenario 3 Model |
| --- | --- | --- | --- |
| Number of neural nodes | 12 | 11 | 11 |
| Activate function | sigmoid | sigmoid | Sigmoid |
| Learning rate | 0.04 | 0.06 | 0.04 |
| Gradient descent function | RMSPropOptimizer | AdamOptimizer | AdamOptimizer |

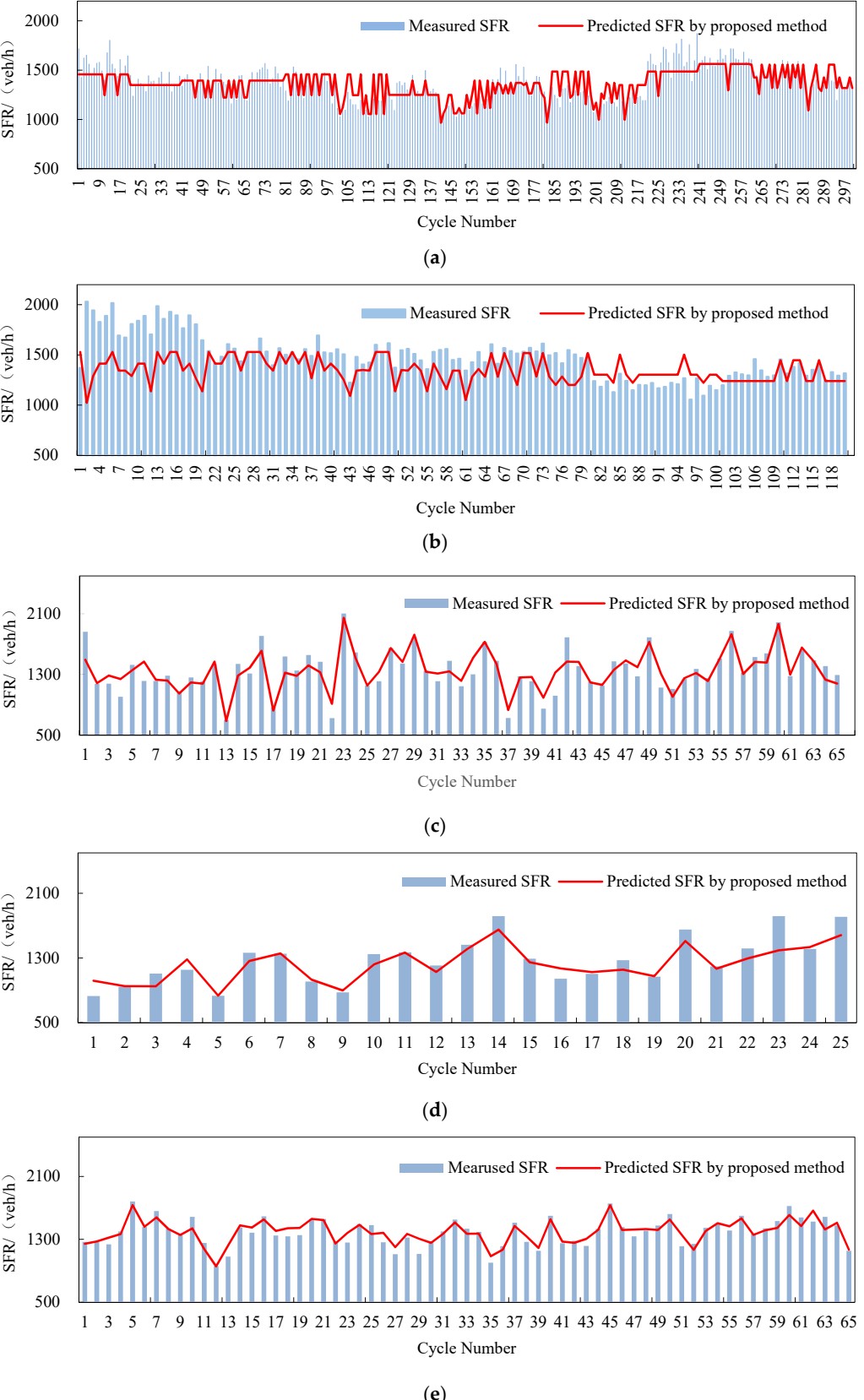

**Figure 7.** *Cont.*

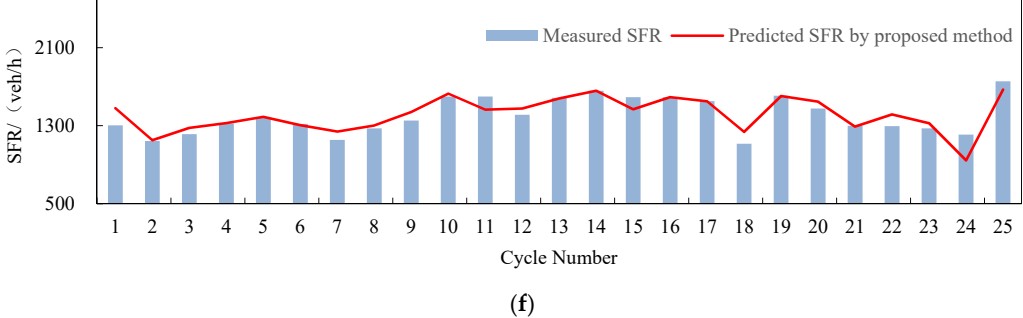

(**f**)

**Figure 7.** Comparison between predicted and measured saturation flow rate (SFR) with training and test sets: (**a**) comparison between predicted and measured results with the training set in scenario 1; (**b**) comparison between predicted and measured results with the test set in scenario 1; (**c**) comparison between predicted and measured results with the training set in scenario 2; (**d**) comparison between predicted and measured results with the test set in scenario 2; (**e**) Comparison between predicted and measured results with training set in scenario 3; (**f**) comparison between predicted and measured results with the test set in scenario 3.

### 5.3. Comparison of Proposed Method and Conventional Method

In order to further verify the effectiveness of the proposed method in this paper, the proposed method was compared with conventional methods which were the HCM method and the statistical method. In scenario 1, the adjacent vehicle lane changes factor was not considered in the HCM method. There were two methods for this factor. Firstly, we should find the relationship between the factor and the saturation flow rate. The adjustment factor was obtained through derivation. Then it was introduced into the HCM multiplication equation. This method was suitable for the simple relationship between one factor and the saturation flow rate. Secondly, the multiple linear regression method was developed. The influencing factors were independent variables. The saturation flow rate was the dependent variable. In general, the linear regression method was used. If there were so many independent variables, the multicollinearity of the model would occur. In this paper, multiple linear regression methods and the HCM method were used as the control group. In the HCM method, two types of parameters need to be determined. One is the base saturation flow rate, and the other is adjustment factors. In order to make the estimated saturation flow rate close to the measured, the base saturation flow rate, lane width adjustment factor and percentage of heavy vehicles adjustment factor were used the recommend values in Chinese National Standard (standard number is GB50647-2011). The left-turn adjustment factor, right-turn adjustment factor and pedestrian-bicycle adjustment factor were recommended in the Chinese National Standard. So they were used in Highway Capacity Manual. For the multiple linear regression method, the three scenarios models were developed with training set data by SPSS software. The calibrated results of the three models were shown in Table 5. The goodness of fit of models was 0.436, 0.355, 0.170, respectively. It was found that the explaining variables (adjustment factors) could not fully explain the changes of explained variables (saturation flow rate).

Due to the interaction between adjustment factors, the HCM method can not express the complex relationship between the factors. As is shown in Figure 8, the saturation flow rates for different lane width and different percentage of heavy vehicles were collected in Beijing. The "m" was the measured saturation flow rates. The "hcm" was the adjusted saturation flow rates with the HCM method. It was found that the changes in adjusted saturation flow rates are different from the measured. In the narrow lanes and high percentage of heavy vehicle scenarios, the measured saturation flow rate decreased rapidly. It was shown that there was an interaction between the lane width and the percentage of heavy vehicles.

<div align="center">**Table 5.** Model parameters and goodness of fit.</div>

| Model | | B [1] | Std. Error [2] | t [3] | Sig. [4] |
|---|---|---|---|---|---|
| Scenario 1 | Constant | 655.694 | 116.511 | 5.628 | 0.000 |
| | PoHV [5] | −96.768 | 16.911 | −5.722 | 0.000 |
| | LW [6] | 243.163 | 35.738 | 6.804 | 0.000 |
| | MTL [7] | −1529.065 | 131.745 | −11.606 | 0.000 |
| | $R^2 = 0.463$ | | | | |
| Scenario 2 | Constant | 1969.191 | 209.556 | 9.397 | 0.000 |
| | PoHV | −121.196 | 241.416 | −0.502 | 0.617 |
| | PoRV [8] | −1079.966 | 244.811 | −4.411 | 0.000 |
| | Pedestrians | −0.218 | 0.080 | −2.736 | 0.008 |
| | Bicycles | −0.309 | 0.202 | −1.531 | 0.131 |
| | $R^2 = 0.355$ | | | | |
| Scenario 3 | Constant | 1617.443 | 87.241 | 18.540 | 0.000 |
| | PoLV [9] | −239.322 | 96.053 | −2.492 | 0.016 |
| | Opposing Vehicles | −0.495 | 0.334 | −1.484 | 0.143 |
| | Pedestrians | 0.196 | 0.109 | 1.803 | 0.076 |
| | Bicycles | −0.223 | 0.114 | −1.945 | 0.056 |
| | $R^2 = 0.170$ | | | | |

[1] B means the coefficient of regression. [2] Std. Error means standard error mean which is the estimated standard deviation of the sample mean. [3] t means t-test statistic value. [4] Sig. means p-value and stands for the significance level. [5] PoHV means the percentage of heavy vehicles. [6] LW means lane width. [7] MTL means multiple through lanes. [8] PoRV means the percentage of right-turn vehicles. [9] PoLV means the percentage of left-turn vehicles.

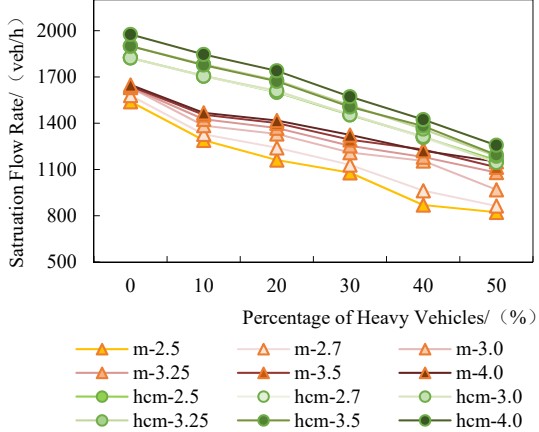

**Figure 8.** Comparison between estimated saturation flow rates (by the Highway Capacity Manual (HCM) method) and measured saturation flow rates.

Based on the test set data of three scenarios, a neural network model which was trained by the training set, a multiple linear regression model which was calibrated by the training set, and an HCM model were used to estimate the saturation flow rate. Each cycle error was calculated and formed the error distribution, as shown in Figure 9. In scenario 1, the mean absolute percentage error (MAPE) of estimated saturation flow rates are 29.30% (HCM model), 12.99% (regression model), 11.23% (proposed model), respectively. In scenario 2, the MAPE of estimated saturation flow rates are 21.60% (HCM model), 14.35% (regression model), 7.02% (proposed model), respectively. In scenario 3, the MAPE of estimated saturation flow rates are 24.53% (HCM model), 11.90% (regression model), 4.70% (proposed model), respectively. It was shown that both the regression model and the proposed model were better than the HCM model. In scenario 1, the error distribution of the proposed model was similar to the regression model. The MAPE of the proposed model was lower than the regression model. Because the traffic scenarios were simple, the saturation flow rates were affected by fewer factors. The linear relationship between the independent and dependent variables was present. The advantage of the neural network model could degenerate into the linear regression model. So the prediction

performance was not significantly different. However, in scenario 2 and 3, the saturation flow rates were affected by many factors. In addition to internal interference (in queue), it would also be interfered with by other external traffic participants (pedestrians and bicycles). The neural network model had obvious advantages, and the accuracy of the neural network model was better than the regression model. It was shown that the neural network model had the advantages in complex scenarios.

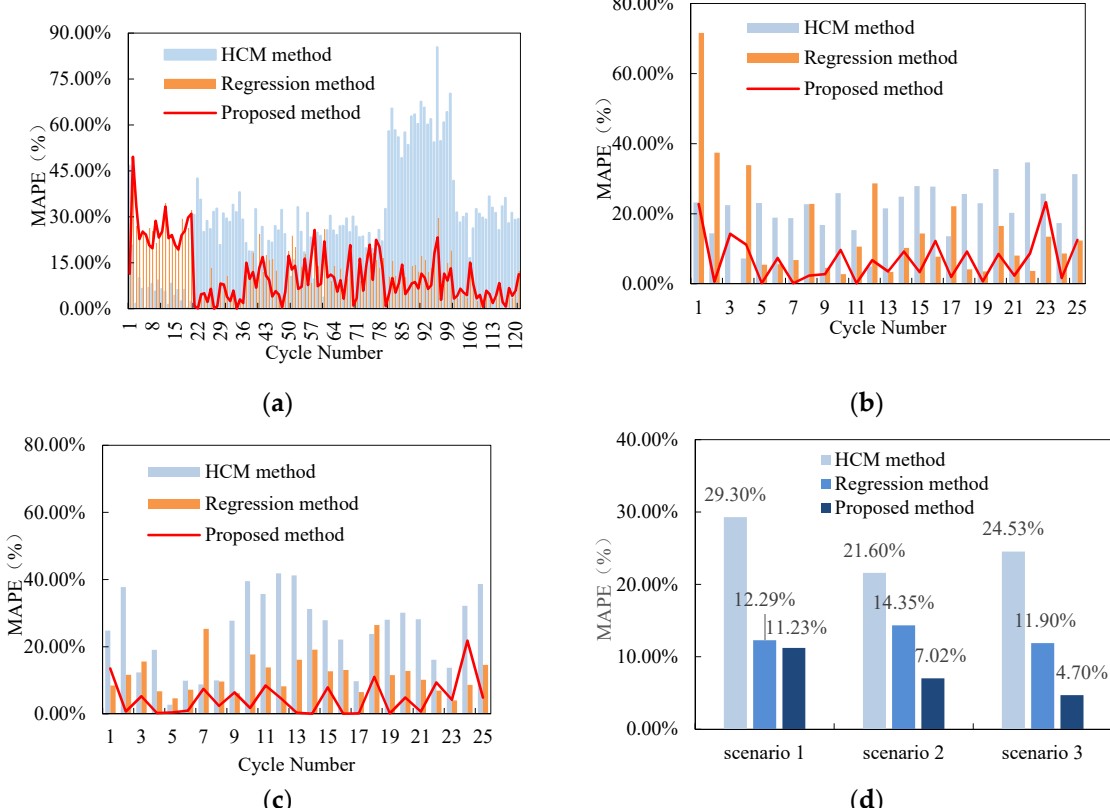

**Figure 9.** Comparison of the predicted accuracy with HCM, regression and Neural Network methods: (**a**) error distributions of three models in scenario 1; (**b**) error distributions of three models in scenario 2; (**c**) error distributions of three models in scenario 3; (**d**) the mean absolute percentage error (MAPE) of estimation saturation flow rates for three models.

*5.4. Potential Applications*

It was demonstrated in an earlier section that the model could estimate the saturation flow rate in the cycle based on the selected variables. The neural network model is an ideal tool to research the effect of some variables on saturation flow rates. When conducting a survey, it is difficult to isolate a variable from the disturbances of other factors. Besides, the neural network model has high flexibility in considering different traffic conditions. In this paper, three models are developed at three traffic scenarios, with different input variables. In fact, the model can accommodate more input variables. The reason for not introducing more variables in this study is that some influencing factors are not considered. During the operation of the intersection, data is collected constantly. The data can cover a variety of traffic conditions. The model can be continuously trained to achieve real intelligence.

In addition, the model can be applied in delay calculation and signal timing. As we know, the delay is the most widely used measure of effectiveness for the level of service analysis of signalized intersections. It is difficult to measure. Researchers have developed delay models for computing. Webster's delay model [27] (see Equation (13)), HCM delay model [3] (see Equation (14)), Akcelik's model [28] (see Equation (15)) and Robertson's model [29] (see Equation (16)) are the most widely adopted.

$$d_p = \frac{c(1-\lambda)^2}{2(1-\lambda x)} + \frac{x^2}{2q(1-x)} - 0.65(\frac{c}{q^2})^{\frac{1}{2}}x^{(2+5\lambda)}, \tag{13}$$

where $d_p$ is the average delay per vehicle on the particular approach, $\lambda = g/c$ is the proportion of the cycle which is effectively green for the phase under consideration, $x = q/\lambda s$ is the degree of saturation, $c$ is the cycle time, $g$ is the effective green time, $q$ is the flow in vehicles per unit time, $s$ is saturation flow.

$$d_1 = \frac{0.5C(1-\frac{q}{c})^2}{1 - [\frac{\min(1,X)g}{c}]}, \tag{14}$$

where $d_1$ is the uniform delay, c is the cycle time, $g$ is the effective green time, $X = q/s$ is the degree of saturation for lane group, $s$ is saturation flow.

$$OD = \frac{cT}{4(X-1) + \sqrt{(X-1)^2 + \frac{12(X-X_0)}{cT}}}, \tag{15}$$

where $OD$ is the overflow delay, $X_0$ is the smallest significant q/s ratio, $s$ is saturation flow.

$$OD = \frac{sT}{c(v-c) + \sqrt{(v-c)^2 + \frac{240v}{T}}}, \tag{16}$$

where $OD$ is the overflow delay, $T$ is the analysis period in minutes, $v$ is volume, $c$ is saturation flow.

There is a large error in the calculation results of the above classic delay model compared with the measured values. Part of the reason is that the saturation flow is introduced to the delay model as a fixed value. In fact, the saturation flow is dynamic changes. If the saturation flow rate is estimated with the proposed model, the accuracy of the delay model will improve. Similarly, in the signal timing process, if the saturation flow rate is introduced signal timing model [30] (see Equation (17)), the timing effect will also be improved.

$$C_0 = \frac{1.5L + 5}{1 - Y}, \tag{17}$$

where $C_0$ is the cycle time, $y$ is the ratio of flow to saturation flow q/s in one phase, $Y$ is the total ratio of each phase, L is the lost time.

## 6. Conclusions

In this paper, the saturation flow rate at the approach of signalized intersections were collected by one cycle. At the same time, the influencing factors were recorded in time series. The saturation flow rates were estimated dynamically in real-time by the neural network. The comparison of the estimated results with HCM, regression, neural network models were shown as follows.

(1) There were many factors that had an effect on the saturation flow rate. They were static factors and dynamic factors. Therefore, the saturation flow rate was not constant, but changed continuously with traffic conditions.

(2) There were basic influencing factors in Highway Capacity Manual. The unique traffic characteristics were not considered in HCM. The complex relationship of influencing factors was difficult to express with the regression model. The neural network can be used to describe the relationship between multiple influencing factors and saturation flow rates.

(3) Compared with the HCM method and Regression method, the more complex traffic scenarios were, the more obvious the advantage of the neural network model. In Scenario 2 and 3, the MAPE of estimated saturation flow rate with a neural network model was 7.02%, 4.07%, respectively.

The proposed method for dynamic estimation of the saturation flow rate was based on the information-rich environment. In the actual application, the model developed and the calibration

process were automatic. Then, connecting with the signal timing system, the goal of refined traffic management was further achieved.

**Author Contributions:** Conceptualization, J.R. and Y.W.; methodology, Y.W.; software, Y.W.; validation, Y.W. and Y.G.; formal analysis, Y.W.; investigation, Y.W., C.Z. and Y.G.; resources, C.Z.; data curation, C.Z.; writing—original draft preparation, Y.W.; writing—review and editing, Y.W. and J.R.; visualization, Y.W.; supervision, J.R.; project administration, C.Z.; funding acquisition, C.Z. All authors have read and agreed to the published version of the manuscript.

**Funding:** This research was funded by the National Natural Science Foundation of China, grant number 5170080357.

**Conflicts of Interest:** The authors declare no conflicts of interest.

## Appendix A

In scenario 1, the weight matrixes $W_{11}$, $W_{12}$, the bias matrixes $b_{11}$, $b_{12}$ were shown as follows.

$$W_{11} = \begin{bmatrix} 1.472 & 0.595 & -2.264 & 1.199 & -1.528 & 0.405 & 0.331 & 0.921 & -1.506 & 0.838 & 0.405 & 0.711 \\ -1.072 & 2.486 & -0.299 & -0.161 & -1.039 & 0.563 & 0.620 & 1.254 & -1.174 & -1.036 & -0.566 & 1.265 \\ -2.351 & 0.481 & -4.884 & 12.228 & -1.522 & -0.304 & 2.132 & 1.054 & -0.587 & 1.659 & -0.554 & 2.303 \end{bmatrix}$$

$$b_{11} = \begin{bmatrix} -0.508 & 1.071 & -0.501 & 0.816 & 1.375 & -0.971 & 0.876 & 1.000 & -0.635 & 1.292 & -0.564 & 1.089 \end{bmatrix}$$

$$W_{12} = \begin{bmatrix} -0.247 & 1.060 & -1.579 & 0.729 & 1.656 & -2.344 & 1.028 & 0.769 & -1.003 & 0.982 & -0.336 & 0.480 \end{bmatrix}$$

$$b_{12} = [0.476]$$

In scenario 2, the weight matrixes $W_{21}$, $W_{22}$, the bias matrixes $b_{21}$, $b_{22}$ were shown as follows.

$$W_{21} = \begin{bmatrix} 1.220 & 0.449 & -1.324 & 0.680 & 0.220 & -0.129 & -0.142 & -0.350 & 1.944 & 1.240 & 0.250 \\ 1.149 & -0.065 & -1.060 & -0.521 & -0.534 & -0.469 & -0.082 & 1.108 & -0.106 & 1.037 & -0.700 \\ 0.586 & 1.140 & 0.496 & 1.061 & -1.235 & 0.866 & -1.502 & -1.755 & -0.459 & -1.446 & 0.108 \\ -0.497 & 0.739 & 0.178 & -1.380 & -1.116 & -1.543 & -1.768 & 0.182 & 0.123 & -0.485 & -1.501 \end{bmatrix}$$

$$b_{21} = \begin{bmatrix} -1.625 & -1.797 & 1.227 & -0.324 & 0.149 & 0.178 & -0.025 & 1.193 & 0.876 & 1.290 & -1.981 \end{bmatrix}$$

$$W_{22} = \begin{bmatrix} -0.261 & 0.039 & -0.076 & -0.515 & 0.251 & -0.596 & 0.939 & -1.329 & -0.386 & 0.438 & -0.514 \end{bmatrix}$$

$$b_{22} = [-0.089]$$

In scenario 3, the weight matrixes $W_{31}$, $W_{32}$, the bias matrixes $b_{31}$, $b_{32}$ were shown as follows.

$$W_{31} = \begin{bmatrix} -0.220 & -1.036 & 1.139 & 1.528 & 1.127 & 0.219 & -0.655 & -0.224 & -0.233 & -0.680 & -1.170 \\ 0.271 & -0.146 & 1.856 & -1.123 & -0.065 & -0.270 & 1.226 & 0.280 & 0.296 & 1.154 & 0.004 \\ -0.076 & -1.707 & -0.946 & -0.074 & -1.434 & 0.075 & 0.054 & -0.079 & -0.086 & -0.736 & 0.395 \\ 0.125 & 1.106 & 0.174 & 1.268 & -0.461 & -0.124 & 0.558 & 0.130 & 0.138 & 0.319 & -0.403 \end{bmatrix}$$

$$b_{31} = \begin{bmatrix} 0.043 & 0.317 & 0.153 & -0.271 & -0.517 & -0.043 & -0.192 & 0.045 & 0.048 & 0.446 & 1.021 \end{bmatrix}$$

$$W_{32} = \begin{bmatrix} -0.387 & 1.302 & 1.240 & -1.497 & -1.691 & 0.384 & -1.216 & -0.398 & -0.419 & -1.626 & -1.076 \end{bmatrix}$$

$$b_{32} = [-0.102]$$

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
