# Peer review of "Dynamic Estimation of Saturation Flow Rate at Information-Rich Signalized Intersections"

_information, doi:10.3390/info11040178_

Round 1

Reviewer 1 Report

In this paper, the authors present a dynamic method for saturation flow rate based on neural networks.  However, I expected to find an in-depth cover of the technical issue and I would like to find a more comprehensive description of their contribution.

In my opinion the sections of introduction and methods are very weak.  Authors describe briefly some related works. In my opinion, they need to provide a more detailed discussion about related works in that area. References are up-to-date. Even though there are several references in the paper, they are not sufficient. The authors need to include more references of magazines and journals. Therefore, they need to discuss the solutions in order to provide a major overview of the proposals and how the research community is facing that problem. I believe this could be done without much difficulty by simply adding some references to the appropriate works. In my opinion, a discussion on these types of works in the appropriate sections can truly give the paper more weight.

The explanation of the methods is very general and it is needed an in-depth description about the neural networks.

The analysis of the results is not explained in detail and I think authors need to provide a more detailed discussion of the results to understand the efficiency of your proposal.

Author Response

Dear editors and reviewers,

All authors thank the editors and reviewers for their time and efforts in reviewing the manuscript. Those comments are valuable and helpful for improving the quality of our work. We have made significant improvements according to the reviewers’ comments. We hope our resubmitted manuscript can meet your journal’s requirements. We appreciate for Editors/Reviewers’ warm work earnestly, and hope that the corrections will meet the approvals of the journal. The manuscript has been resubmitted to your journal. We are looking forward to your positive response. Our detailed responses to the reviewers’ comments are attached below.

Thank you

Best regards.

Reviewer 2 Report

The authors did a good job of describing the limitations of the HCM methodology for estimating saturation flow rate, which is based mainly on static and empirical data. The authors' attempt to incorporate dynamic and real-time data is beneficial, especially in determining saturation flow-rate for operational analysis of an intersection.

The authors, and other researchers in the past, have shown that neural network in traffic analysis is a promising approach when dealing with complex and non-linear variables that characterize signalized intersections.  

Lines 55 and 59: filed should be field

Line 243: 5 mim should be 5 min

Line 344: date should be data

Line 298-300: Description of the intersection in Scenario 1 is unclear and needs to be revisited by the authors.  There was no mention of left turn lanes, and unclear description of the number of lanes in various approaches. 

Lines 357-258: The minimum saturation flow rate is 967.  The maximum saturation flow rate is 967.  (The maximum saturation flow rate on the table is 2030.62).

Author Response

(The authors gave the same response as above.)

Reviewer 3 Report

Interesting subject and application.

Well-structured paper and clear presentation of the methods and results.

The introduction could be enriched with relevant applications from the literature.

Sources needed for lines 134-137, 147 and figures 2, 3.

The last paragraph of section 2.1 along with figure 1 present results. This part should either be embedded in the methodology or moved to the results.

Furthermore, extensive language editing and proof reading is required to improve readability and quality of the text. For example, lines 82-84 need to be rechecked and also all the cases using "we". The same applies to 397-398 with "you".

Line 344 "date" to be checked.

Figure 6 is not necessary. 

Figure 7 is not clear.

Author Response

(The authors gave the same response as above.)
